# Super-elasticity at 4 K of covalently crosslinked polyimide aerogels with negative Poisson's ratio

Yang Cheng[1,2], Xiang Zhang [3], Yixiu Qin[4], Pei Dong[5], Wei Yao[1,2], John Matz[5], Pulickel M. Ajayan [3], Jianfeng Shen [1✉] & Mingxin Ye[1✉]

The deep cryogenic temperatures encountered in aerospace present significant challenges for the performance of elastic materials in spacecraft and related apparatus. Reported elastic carbon or ceramic aerogels overcome the low-temperature brittleness in conventional elastic polymers. However, complicated fabrication process and high costs greatly limited their applications. In this work, super-elasticity at a deep cryogenic temperature of covalently crosslinked polyimide (PI) aerogels is achieved based on scalable and low-cost directional dimethyl sulfoxide crystals assisted freeze gelation and freeze-drying strategy. The covalently crosslinked chemical structure, cellular architecture, negative Poisson's ratio ($-0.2$), low volume shrinkage (3.1%), and ultralow density (6.1 mg/cm$^3$) endow the PI aerogels with an elastic compressive strain up to 99% even in liquid helium (4 K), almost zero loss of resilience after dramatic thermal shocks ($\Delta T = 569$ K), and fatigue resistance over 5000 times compressive cycles. This work provides a new pathway for constructing polymer-based materials with super-elasticity at deep cryogenic temperature, demonstrating much promise for extensive applications in ongoing and near-future aerospace exploration.

[1] Institute of Special materials and Technology, Fudan University, Shanghai, China. [2] Department of Materials Science, Fudan University, Shanghai, China. [3] Department of Materials Science and Nanoengineering, Rice University, Houston, TX, USA. [4] State Key Laboratory of Molecular Engineering of Polymers, Fudan University, Shanghai, China. [5] Department of Mechanical Engineering, George Mason University, Fairfax, VA, USA. ✉email: jfshen@fudan.edu.cn; mxye@fudan.edu.cn

n the field of aerospace exploration, spacecraft and supporting apparatus often suffer from the impact of deep cryogenic environments. For instance, the lowest temperature on the surface of Mars is 130–140 K[1], while the temperature is as low as 50 K on the moon's poles[2]. The Space Shuttle Challenger event shocked the whole world as exploding within 73 s after its takeoff due to the elastic failure of rubber O-ring at low temperature, indicating the vitally essential role of elastic materials resistant to the cryogenic environment for aerospace exploration.

Unfortunately, most of the conventional intrinsic elastic materials, such as thermoplastic elastomers, natural and synthetic rubbers, generally tend to lose their intrinsic elasticity in deep cryogenic environments[3,4]. To address this problem, structurally elastic aerogels, mainly based on carbon[5] and ceramics[6], have captured researchers' attention due to their satisfactory elasticity from three-dimensional (3D) network architectures and excellent resistance to deep cryogenic conditions. For instance, graphene-coated carbon nanotubes (CNTs) aerogels[7–10] and carbon nanofibers (CNFs) aerogels[11] can bear the compressive strain of 50% to 90% at 173 K. Notably, Chen et al. created graphene aerogels with satisfying recoverability under 98% compressive strain at 77 K[12] or resilience under 90% strain at the deep cryogenic temperature of 4 K[13]. Moreover, ceramic aerogels of BN nanoribbon and nanofibrous $SiO_2$-based composites are also in possession of compressive super-elasticity at 77 K[14–17]. These newly emerged carbon and ceramic aerogels make significant promotion for the development of elastic materials in deep cryogenic environments, while their complex fabrication process and high cost still raise misgivings.

In this regard, with easy processability and low-cost fabrication, it will be much more intriguing if special polymers can be synthesized and achieve super-elasticity in deep cryogenic environments. Wang et al recently demonstrated a polymeric aerogel composed of low-cost chitosan and melamine-formaldehyde resin, with super-elasticity at liquid nitrogen temperature (77 K), which opens up a new avenue for further development of elastic polymeric materials resistance to deep cryogenic temperature[18]. Among polymeric materials, polyimide (PI) with remarkable resistance to extreme conditions (fire, radiation, chemical corrosion, low and high temperature, etc.) is considered to be potentially ideal candidates for elastic materials applied at deep cryogenic temperatures[19–22]. Generally, freeze-casting techniques based on water-soluble PI precursors of poly (amic acid) ammonium salt (PAAS) are widely applied in the fabrication of elastic PI aerogels[23,24]. Based on this strategy, various elastic PI aerogels composited with CNTs[25,26], graphene[27,28], silica[29], MXene[30,31], and nanofibers[32,33] have been produced, endowing PI aerogels with such promising applications as electromagnetic shielding, oil-water separation, pressure sensors, etc. Unfortunately, the thermal imidization after freeze-drying in the above strategy inevitably causes large shrinkage up to 40%, greatly impairing the compressibility of elastic PI aerogels[34]. Furthermore, the decomposition of PAAS in water cannot be completely avoided as a result of imperfect salinization of poly (amic acid) (PAA), leading to weak resilience of elastic PI aerogels because of low molecular weight. The recently emerged electro-spun nanofibrous PI aerogels provide an effective pathway to avoid the large shrinkage and decomposition of PAAS in water[35–37], but the incorporation of the electro-spun process complicates the whole fabrication process and increases the costs.

In this study, we proposed a directional dimethyl sulfoxide crystal assisted freeze gelation and freeze-drying (DMSO-FGFD) strategy to construct covalently crosslinked PI aerogels with super-elasticity at deep cryogenic temperatures even down to 4 K. Chemical imidization without water can be realized to transform PAA into PI oligomers at room temperature in this strategy, thus resulting in low volume shrinkage of 3.1% and density of 6.1 mg/cm³, which are far superior to elastic PI aerogels from conventional thermal imidization. Meanwhile, innovative mold design and temperature adjustment endow the obtained PI aerogels with the radially distributed cellular structure to realize negative Poisson's ratio (NPR) behavior. Thanks to the covalently crosslinked chemical structure, favorable NPR behavior, low shrinkage, and density, the prepared PI aerogels are endowed with fully reversible super-elastic behavior of up to 90% strain, satisfying stability of compressive cycles over 5000 times. Furthermore, the fantastic super-elasticity and fatigue resistance is proved to be temperature invariant over the wide temperature range from 4 K to 573 K, and almost zero loss of resilience is observed even after dramatic thermal shocks ($\Delta T = 569$ K).

## Results

**Fabrication of PI aerogels.** Fabrication of covalently crosslinked PI aerogels with super-elasticity was demonstrated in Fig. 1. Firstly, PI oligomers end-capped with anhydride were obtained through chemical imidization at room temperature by adding acetic anhydride and triethylamine into PAA precursors which were synthesized from 4,4'-oxydianiline (ODA) and 4,4'-oxy-diphthalic anhydrides (ODPA) in DMSO solvent (Supplementary Fig. 1). Subsequently, a directional freeze gelation process was carried out by adding the DMSO solution containing PI oligomers and 1,3,5-triaminophenoxybenzene (TAB) crosslinkers into a predesigned model subjected to a programmable temperature gradient. At the initial freeze gelation stage, the DMSO crystals grew horizontally from the periphery to the center, resulting in radially distributed crystals due to a predesigned model and temperature adjustment. After that, the covalently crosslinked PI was formed between vertically grown DMSO crystals. Finally, 3D honeycombed PI aerogels with the radially distributed cellular structure were obtained after freeze-drying to remove DMSO and thermal treatment to transform residual PAA units into the PI. The crosslinking degree of the obtained PI aerogels could be controlled by adjusting the molar ratio of ODPA, ODA, and TAB, which is described detailly in the section "Methods". The obtained crosslinked PI aerogel is marked as PI-10, PI-20, PI-30, PI-40 when the initial PI oligomers maintain the polymerization degrees of 10, 20, 30, and 40, while PI-L is corresponding to PI aerogels prepared without crosslinker. Theoretically, short polymer chains are easier to orient under shear stress due to fewer entanglements between them, resulting in a faster shear-thinning effect and lower shearing viscosity. As shown in Supplementary Fig. 2, the shearing viscosity curves of the PI oligomers solution demonstrated an increasingly rapid decrease of viscosity in the shearing thinning region from PI-L to PI-10, as well as progressively lower constant shearing viscosity in the constant viscosity region, illustrating progressively shorter polymer chains from PI-L to PI-10. The molecular weights of oligomers were further investigated with gel permeation chromatography (GPC), and the results were shown in Supplementary Fig. 3. It demonstrates that the polymerization degree ($\bar{X}_n$) of PI-L can reach 98, and $\bar{X}_n$ of the other oligomers are consistent with the experimental design (10, 20, 30, 40), which also accords with the viscosity test results. More TAB crosslinkers are added into the solution containing shorter PI oligomers to ensure complete reaction of the anhydride groups terminated PI oligomers, resulting in higher crosslinking degrees in the final PI aerogels in accordance with the initial design. Benefiting from the facile process and low cost of raw materials, bulk PI aerogels with a volume of 300 cm³ and diverse shapes have been prepared, demonstrating the feasibility of large-scale fabrication based on this creative DMSO-FGFD strategy.

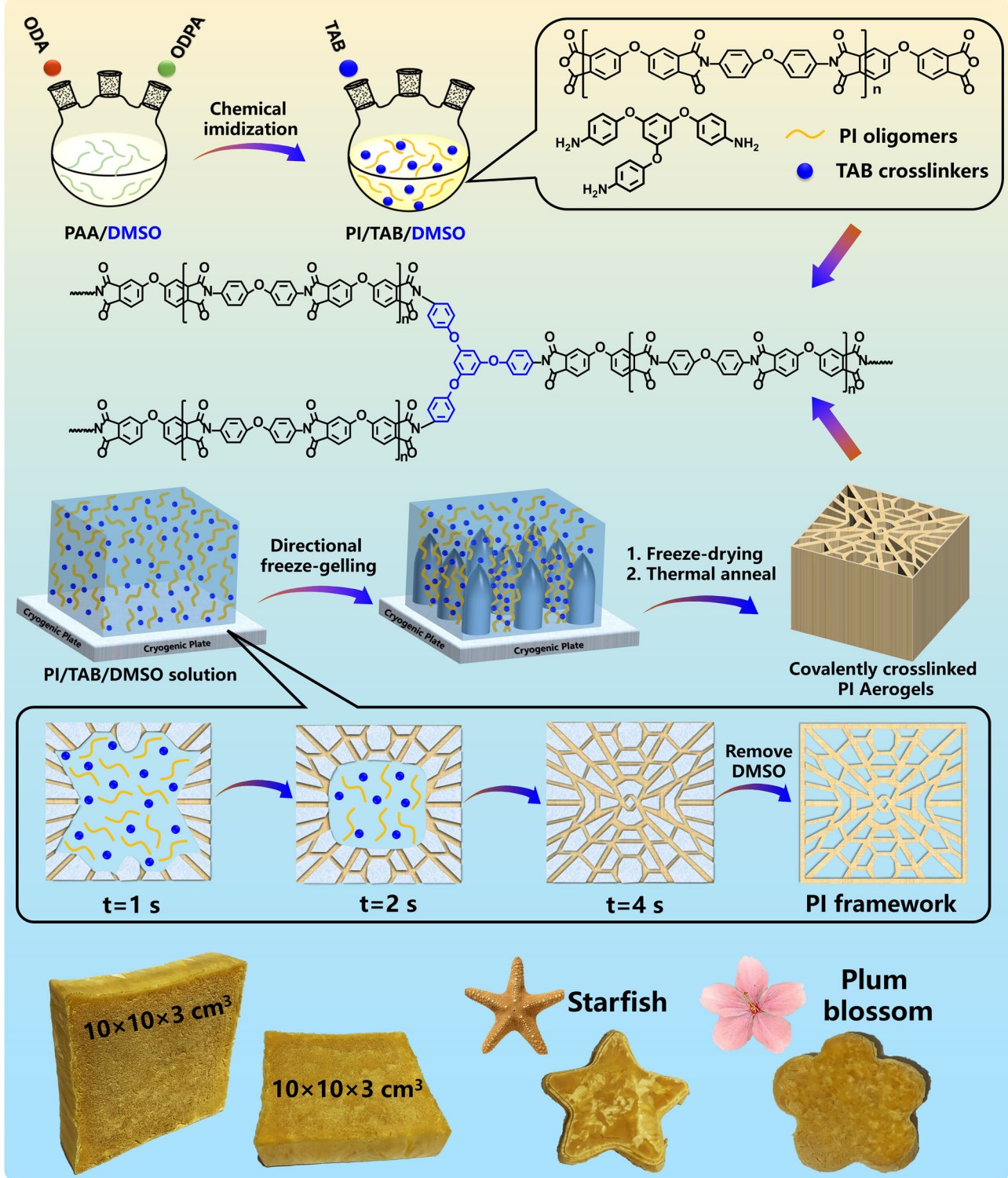

**Fig. 1 Schematic illustration of constructing super-elastic PI aerogels.** Design and synthesis of PI aerogels with covalent crosslinking, radially distributed cellular structure, and diversely shaped bulk aerogels.

**Investigation of DMSO-FGFD process**. The DMSO-FGFD process was further investigated in depth. To explore the process of gelling interaction between PI oligomers and TAB crosslinkers, the rheological behavior of PI/TAB/DMSO mixtures with a solid content of 6 wt% was observed on a rotational rheometer as shown in Fig. 2a. Under a constant shearing rate, PI-L/DMSO without TAB crosslinkers shows a relatively high zero shear viscosity but stays constant. In contrast, the viscosities of the PI/TAB/DMSO mixtures display sharp increase at the initial stage, which demonstrates high reactivity between PI oligomers and TAB to form covalently crosslinked PI. Besides, from PI-40 to PI-10, the viscosity rising rate and final viscosity tend to increase, which should be mainly attributed to a higher content of reactive groups in the solution with shorter oligomers. Figure 2b

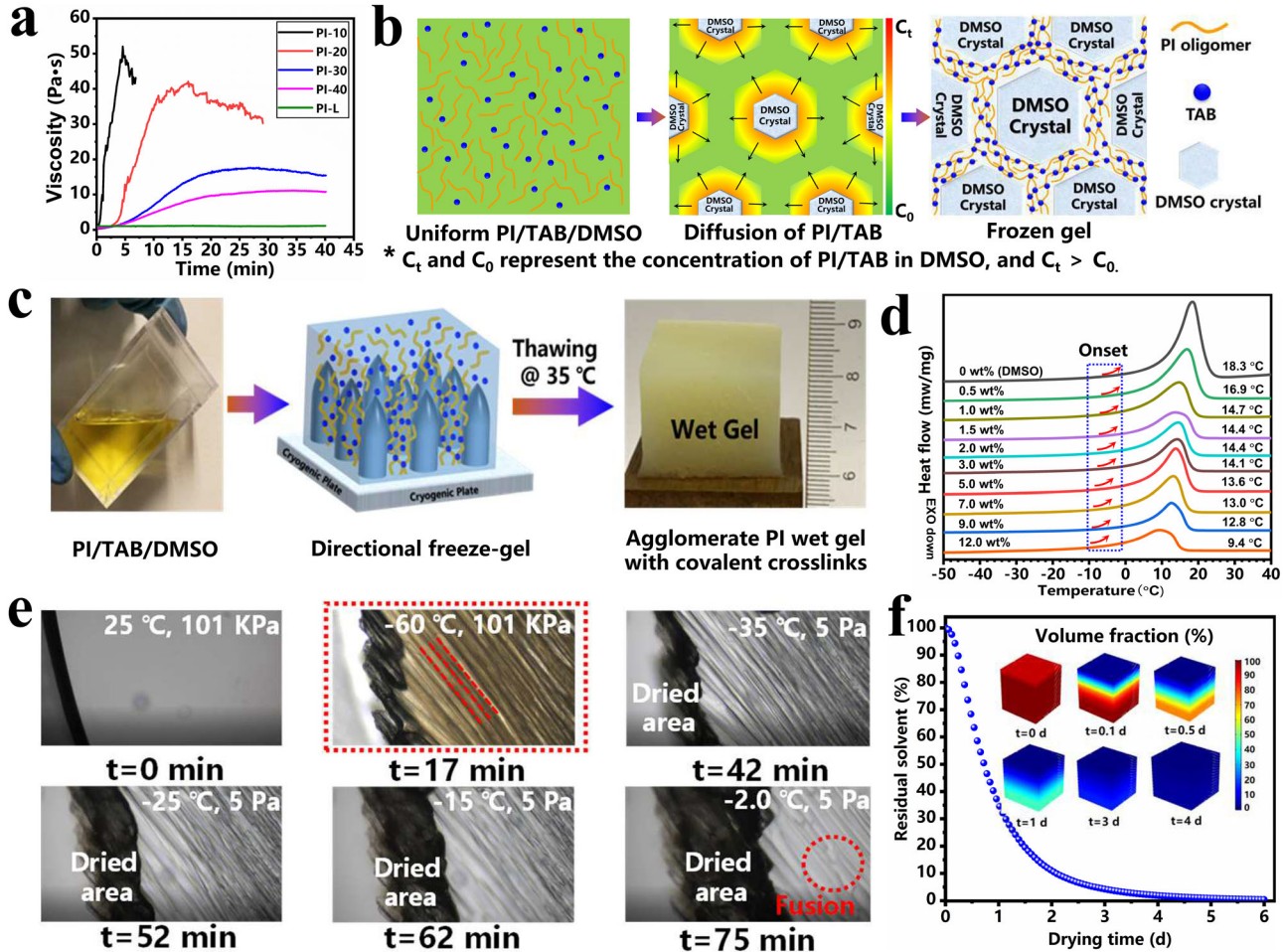

**Fig. 2 Fabrication process of covalently cross-linked PI aerogel with DMSO solvent. a** Shearing viscosity curves of 6 wt% PI oligomers solutions with different contents of TAB cross-linker under a constant shearing rate. **b** Mechanism of the directional freeze gelation process. **c** Optical images after thawing frozen gel of PI/TAB/DMSO with a solid content of 0.5 wt%. **d** DSC cures of DMSO solvent containing different content of PI oligomers with polymerization degree of 10. **e** Microscopy images of the pre-freezing and freeze-drying process from the microscopy system of vacuum freeze-drying based on PIM-10 with a 0.5 wt% solid content in DMSO solvent. **f** Simulation of residual DMSO during the freeze-drying process by finite element analysis.

illustrates the mechanism of the freeze gelation process. PI oligomers and TAB showed a very low reaction rate and almost stayed uniform in a dilute solution. Upon freezing, phase separation took place, and PI oligomers with TAB were expelled to the boundaries of the DMSO crystals because of the volume exclusion effect, resulting in an increase in the localized concentration of PI/TAB around vertical DMSO crystals[38]. With the continuing growth of DMSO crystals, PI oligomers and TAB diffused to diluted area driven by the concentration gradient, thus forming a high concentration area between DMSO crystals, which significantly promoted the reactivity between PI oligomers and TAB to form the crosslinked PI networks. As a result, an aniso-tropic frozen gel composed of covalently crosslinked PI and DMSO crystals was obtained. After thawing at 35 °C, the frozen gel with TAB transformed into an agglomerate wet gel (Fig. 2c), verifying the formation of PI with covalent crosslinking in the freeze gelation process, while the frozen gel without TAB returned to a flowing state in contrast (Supplementary Fig. 4).

Different from the process using water as the solvent, the freeze-drying process with DMSO has rarely been investigated previously. It is generally accepted that the stabilization of frozen monoliths is crucial to obtain a satisfactory 3D architecture in a freeze-drying process. The Differential Scanning Calorimetry (DSC) curves demonstrated the melting temperature range of

DMSO crystals containing different amounts of PI oligomers with a polymerization degree of 10 (Fig. 2d). As the contents of PI oligomers in DMSO reduced from 12 wt% to 0.5 wt%, the melting points (peak value) tend to increase from 9.4 to 16.9 °C closing to the melting point of pure DMSO (18.3 °C). The onset melting points of DMSO solution with different concentrations ranging from −10 to 0 °C, which determine the upper limit of the freeze-drying temperature at the initial stage. Furthermore, DMSO solvent containing linear PI with a solid content of 0.5 wt% was chosen to in-situ observe the formation of DMSO crystals in the pre-freeze and analyze the temperature range of vacuum drying by the vacuum freeze-drying microscopy system. As shown in Fig. 2e, when cooled down from 25 °C to −60 °C at ambient pressure, parallel DMSO crystals come into being as templates to push PI chains to aggregate between them, demonstrating that the DMSO solvent was beneficial to prepare PI aerogels with well-organized directional frameworks. The dried area began to enlarge as the temperature rising from −60 to −35 °C at vacuum (5 Pa), and then crystal fusion was observed as the temperature reaching −2.0 °C, which was in accord with the DSC results. It illustrated that the temperature range of the vacuum freeze-drying process is −35 to −2 °C. As a high boiling point solvent (189 °C) with low saturated vapor pressure, DMSO is not easy to be dried. According to simulated results based on finite element

analysis, 4 days are necessary to completely finish the vacuum drying process, which is in line with the experimental results (Fig. 2f).

**Structure and morphology.** The PI aerogels freeze-dried from 0.5 wt% PI/TAB/DMSO solution has been chosen to investigate the structure and morphology of PI aerogels with NPR behavior. The chemical structure of PI aerogels has been characterized by Fourier Transform Infrared Spectroscopy-Attenuated Total Reflection (FTIR-ATR). As seen in Supplementary Fig. 5, the spectra exhibit typical characteristic peaks of imide structure at 1776 cm$^{-1}$ (imide C=O asymmetric stretching), 1714 cm$^{-1}$ (imide C=O symmetric stretching), 1371 cm$^{-1}$ (C–N stretching vibration), 1014 cm$^{-1}$, and 742 cm$^{-1}$ (C–N–C stretching vibration). DSC was further taken to analyze the crosslinking structure in PI aerogels. According to the research of Loshaek[4], $T_g$ of a polymer shows a positive correlation with its crosslinking degree as shown in the following equation.

$$T_g = K_x\rho + T_g(\infty) - \frac{K}{M} \quad (1)$$

where $T_g$ and $T_g$ ($\infty$) are the glass transition temperatures of crosslinking polymer and linear polymer, respectively, $\rho$ is the crosslinking degree, $M$ represents the molecular weight, $K_x$ and $K$ are constants. Obviously, the $T_g$ of PI aerogels exhibits a remarkable upward trend from 252 °C of PI-L to 285 °C of PI-10, clearly demonstrating the increase of crosslinking degrees (Fig. 3a).

PI aerogels produced by freeze-casting based on PAAS precursor usually suffer from severe volume shrinkage due to thermal stress shock and free volume reduction during the thermal imidization process at 200–300 °C, which greatly hinder its practical applications. In this work, benefiting from the good solubility of DMSO, the chemical imidization process and a covalently crosslinked structure were achieved simultaneously to fabricate PI aerogels, which synergistically mitigate the volume shrinkage of the obtained aerogels. As shown in Fig. 3b, PI aerogels made by chemical imidization with DMSO display volume shrinkage less than 7.3%, which is much superior to those of 19.5–25.3% from thermal imidization. In addition, with the increasing of crosslinking degrees, shrinkage tends to be inhibited gradually whether by chemical imidization or thermal imidization. Notably, the volume shrinkage of PI-10 aerogels by chemical imidization can be as low as 3.1%, which is far superior to any reported PI aerogels[23,32,39]. It can be attributed that chemical imidization can transform PAA into PI before thermal annealing, thus avoiding the decrease of free volume in thermal imidization, which is proved by molecular dynamic simulation in Supplementary Fig. 6. Apart from that, the covalently crosslinked structure usually endows PI aerogels with much better thermal resistance and mechanical properties, which could inhibit the structural damage by thermal stress shock in thermal annealing at high temperatures. Shrinkage of elastic PI aerogels can vary with different chemical structures and constitutions in thermal imidization, while the chemical imidization based on the DMSO-FGFD method should be universal for most elastic PI aerogels to restrain shrinkage effectively.

Little volume shrinkage is the premise for ultralow density, high porosity, and well-organized structure in PI aerogels. As shown in Fig. 3c, a 3 cm × 3 cm × 3 cm bulk aerogel of PI-10 weighs only 164.2 mg, indicating its density is as low as 6.1 mg/cm$^3$, while the corresponding porosity is up to 99.57%. As demonstrated by scanning electron microscope (SEM) images (Fig. 3d) viewing along $z$ direction ($z$ view) and viewing along $x$ or $y$ direction ($x$, $y$ view), PI-10 aerogel consists of parallel hollow tubes with a pore size of 200–300 μm and a wall thickness of

~2 μm, which is the result of evaporation of DMSO crystals in vacuum freeze-dryer. In contrast, the PI aerogel fabricated by freeze-casting method based on 1 wt% PAAS aqueous solution exhibits the unregular porous microstructure (Supplementary Fig. 7). In addition, in view of the effect of mold inner walls and the lower freeze temperature at the mold bottom, the morphology of skin layers at the periphery and bottom of PI aerogel has also been investigated in Supplementary Fig. 8. The skin layer around the aerogel is a well-organized channel structure, which is similar to the major architecture, revealing that the skin layer should be a horizontally periodic spread of the unidirectional architectures, but the bottom skin layer displays smaller sized pores (10–20 μm). Such ultralow density, high porosity, and well-organized morphology mainly benefit from the minimal shrinkage and ordered DMSO crystals formed in the pre-freezing process. Apart from that, by fine-tuning the solid contents of PI/TAB/DMSO mixtures, the density of final PI aerogels can be tuned from 6.1 mg/cm$^3$ to 52.5 mg/cm$^3$, corresponding to a porosity change from 99.57% to 99.29% (Supplementary Fig. 9). Thus, with this pioneering DMSO-FGFD process, the density, porosity, and wall thickness can be adjusted flexibly according to actual practical requirements.

NPR behavior was observed in the obtained PI aerogels due to innovative mold design and temperature adjustment[40–42]. With the help of finite element simulation, a radial temperature distribution (Fig. 3e) was achieved at the initial stage of freeze gelation through a design with a slightly sunken center on the outer bottom of the mold (Supplementary Fig. 10). The contrivable temperature distribution was capable of controlling the growing direction of DMSO crystals from the periphery to the center on the inner bottom (Supplementary Movie 1), resulting in a radially distributed cellular structure of PI aerogels (Fig. 3f). According to simulation results, the special structure-network reveals a hyperbolic-patterned deformation under compression, depicting obvious NPR behavior (Fig. 3g). Figure 3h presents the real longitudinal ($\varepsilon_y$) and transverse ($\varepsilon_x$) strain evolution of PI aerogels under loading, demonstrating hyperbolic-patterned shrinkage in the macroscopic configuration in a transverse direction under longitudinal compression. $\varepsilon_y$ decreases from 0% to −41.7% accompanied by $\varepsilon_x$ decreasing from 0% to −8.3% indicating significant NPR behavior from 0 to −0.20 calculated by $\nu = -\varepsilon_x/\varepsilon_y$. The favorable NPR behavior is beneficial to the super-elasticity of PI aerogels due to a wide distribution of compressive strain and better dissipation of impact energy[40,43].

**Evaluation of super-elastic performance.** Benefitting from the ultra-low density, radially distributed cellular structure with NPR behavior and enhanced crosslinking networks of PI chains, the acquired PI aerogels display anisotropic mechanical performance, such as high stiffness along channel direction and ultra-high flexibility on vertical channel direction. Interestingly, the bulk PI-10 aerogel with a low density of 6.1 mg/cm$^3$ is able to bear 2000 times of its own weight (Supplementary Fig. 11) along channel direction, which clearly reveals its strong stiffness. Besides, as shown in Fig. 4a, on vertical channel direction, they are capable to recover under 180° bending several times (Supplementary Movie 2) and 99% compressive strain (Supplementary Movie 3), indicating amazing flexibility and super-elasticity. The effect of crosslinking degree on compressive stress of PI aerogels was further investigated. As shown in Fig. 4b, all prepared PI aerogels with ultra-low densities of 6.1 to 8.0 mg/cm$^3$ (Supplementary Fig. 9) can recover under 70% compressive strain vertical channel direction and exhibit a growing tendency of stress with increasing crosslinking degree. Covalently crosslinked structure endows

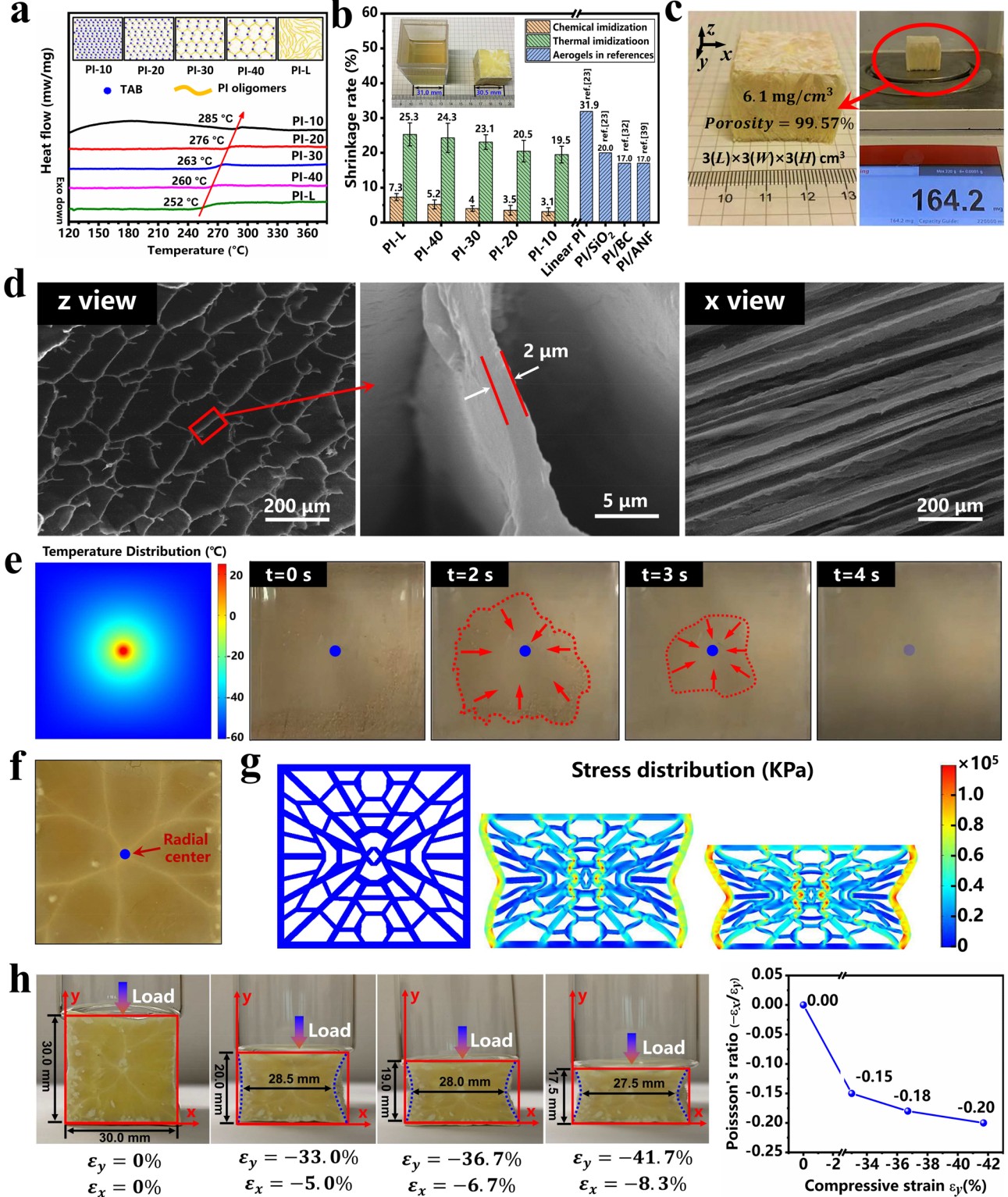

**Fig. 3 Structure and morphology of PI aerogels. a** DSC curves of PI aerogels. **b** Average shrinkage rates of PI aerogels from chemical imidization, thermal imidization, and related references, 5 parallel tests were performed on each series of samples. **c** Optical images of the volume and weight of a typical PI aerogel. **d** SEM images viewing along z-direction (z view) and along x-direction (x view) of anisotropic PI-10 aerogels. **e** Temperature distribution and growing direction of DMSO crystals at the bottom of the mold. **f** Overview of radial distributed morphology in PI-10 aerogels. **g** Simulated NPR behavior of PI aerogels. **h** Sequential optical images showing NPR under compression vertical channel direction.

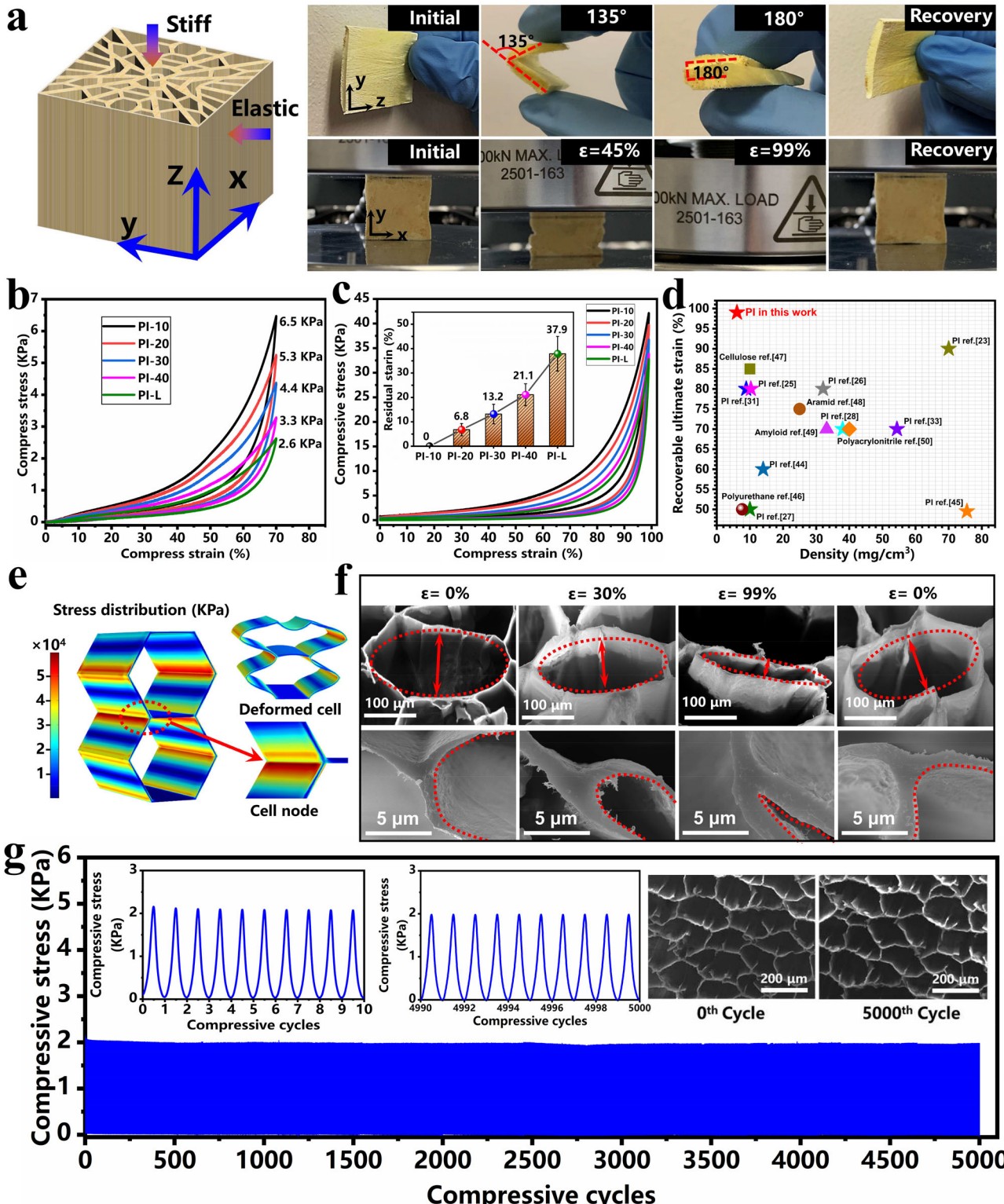

**Fig. 4 Mechanical performance of PI aerogels. a** Optical image of a PI aerogel's recovery after 180° bending and 99% compression. **b** Compressive stress-strain curves with 70% strain of PI aerogels with different cross-linking degrees. **c** Compressive stress-strain curves with 99% strain and residual strain of PI aerogels with different cross-linking degrees. **d** Comparison of ultimate recoverable strains and densities of PI-10 aerogels with reported polymeric aerogels. **e** Simulated results of local stress distribution in cellular PI aerogels. **f** In-situ sequential SEM images of microstructure in PI-10 aerogels with different strains. **g** Fatigue test curves and SEM images of PI-10 aerogels before and after fatigue test of 5000 cycles.

PI-10 aerogels with a compressive stress of 6.5 kPa under 70% strain in vertical channel direction, which is 2.5 times that of PI-L without crosslinking.

The compressibility and elasticity vertical channel direction of PI aerogels have been further evaluated under an ultimate strain of 99%. As shown in Fig. 4c, all PI aerogels with different cross-linking degrees can be compressed to 99% due to their extraordinary flexibility and ultra-low densities of 6.1 to 8.0 mg/cm$^3$. However, in contrast, to complete recovery under 70% compressive strain, PI-L suffers from serious plastic deformation with a residual strain of 37.9% after 99% compression (Supplementary Fig. 12). With the incorporation of a covalently crosslinked structure, elastic recovery has been improved, while the residual strain gradually decreases with the increase of crosslinking degree. PI-10 aerogels are capable of springing back to their original shape, revealing excellent super-elastic performance. In addition, in view of the effect of skin layers at the periphery and bottom of PI aerogel, a comparison investigation has been carried out in Supplementary Fig. 13. It demonstrated minor improvement of compressive stress but little impact on the ultimate resilient strain of skin layers. When compared with reported PI aerogels and other polymeric aerogels, PI-10 aerogels display much lower density but far superior elastic properties (Fig. 4d)[23,25–28,31,33,44–50].

The highly recoverable compressibility vertical channel direction of PI-10 aerogels can be mainly attributed to the enhanced mechanical properties because of their covalently crosslinked structure and NPR behavior via their radially distributed cellular structure. In order to deeply investigate the promotion of enhanced constituent PI to the super-elasticity of PI-10 aerogels, structural variations in compression have been analyzed in detail. PI aerogels in this work are assembled with thin PI walls connected by cell nodes that are the main supporting parts of the framework. Figure 4e demonstrates the simulated results of stress distribution and deformation of the cellular structure under compression based on two contiguous honeycomb-configured models with a commonly used connectivity of three in single nodes[13,33,51]. While the cellular structure is bearing compressive stress, cell units are approaching each other gradually along the compressive direction, forcing the cell nodes to become stress-concentrated regions which determine the recoverability of aerogels under large compressive strains. To verify the above simulations, Supplementary Fig. 14, and Fig. 4f, respectively, present the overview and local structural evolution of PI-10 aerogels during 99% compression-decompression by in-situ SEM observations. Obviously, the cell units undergo large pressing flat accompanied with a distinct angular variation of cell nodes under 99% compressive strain, which recovers its original shape without any structural damage after the release of stress due to the enhanced mechanical properties from the covalently crosslinked structure of PI aerogels. In contrast, damage to the cell nodes in PI-L with relatively lower strength is observed after 99% compression, resulting in obvious plastic deformation (Supplementary Fig. 15). Apart from the enhanced constituent PI in aerogels, framework structures with NPR behavior endow PI-10 aerogels with hyperbolic-patterned deformation under compression. The structural variations possess wide distributions of compressive strain and better dissipation of impact energy, mitigating structural damage to ensure perfect recoverability during high compression. Under the synergistic effect of covalent crosslinking and NPR behavior, PI-10 obtained much stronger mechanical properties and better dissipation of impact energy so that the cell nodes can stay intact even under 99% compressive strain.

Fatigue resistance and adjustable mechanical properties are two vital factors for aerospace materials. A cyclic compression vertical channel direction with a sinusoidal frequency of 1 Hz test was carried out to estimate the mechanical durability of PI-10 aerogels with a density of 6.1 mg/cm$^3$. Interestingly, there was no significant decrease in compressive stress or cracking failure in the cell structure, even after 5000 compression-decompression cycles, indicating excellent long-term stability of PI-10 aerogels (Fig. 4g). In addition, by tailoring the solid content of PI/TAB/DMSO mixtures from 0.5 to 3.0 wt%, the wall thickness of PI-10 aerogels varies from 2 to 10 μm (Supplementary Fig. 16a), corresponding to a stress variation of 6.7 to 26.1 KPa at 70% compressive strain (Supplementary Fig. 16b). It reveals that a higher PI concentration in the DMSO solution results in thicker walls and more robust mechanical properties, demonstrating manipulatable structural and mechanical performances.

**Evaluation of super-elasticity at deep cryogenic temperature.** The super-elasticity of PI-10 aerogel was further evaluated in a gradually freezing environment from 573 K to 4 K (Fig. 5a). PI aerogels exhibit rising glass transition temperature (Fig. 3a) and thermal decomposition temperature with the increase of cross-linking degree, and the $T_{d5}$ (temperature of 95% residual weight) of PI-10 aerogels is up to 539 °C which is 19 °C higher than that of PI-L aerogels without crosslinking (Supplementary Fig. 17). Benefiting from the enhanced thermal resistance, PI-10 aerogel is able to completely recover to its original shape after suffering from large compressive deformation at 298 K and 573 K. Furthermore, the PI-10 aerogel was soaked in liquid $N_2$ (77 K). Generally, most polymeric materials become hard and brittle under such circumstances. In contrast, PI-10 aerogel could be circularly compressed with 90% strain several times and still perfectly recover without plastic deformation. Moreover, the elastic behavior of PI-10 aerogels was further investigated by a single uniaxial compress-release operation in liquid helium (4 K) using a customized apparatus (Supplementary Fig. 18). Even at such a deep cryogenic temperature, PI-10 aerogel still possesses excellent resilience after repeated compression up to 90% strain (Supplementary Movie 4). To the best of our knowledge, such remarkable and macroscopic temperature-invariant super-elasticity performances, even down to deep cryogenic temperature, have never been reported for any polymeric materials. In addition, Fig. 5b presents the stress-strain curves of PI-10 aerogels treated in different temperatures (573 K, 298 K, 77 K, and 4 K) for 3 min and then taken out to compressive tests immediately. Note that the stress-strain curves of PI-10 aerogels, which were treated at 4 K, 77 K, and 573 K almost overlap with the curves of aerogels treated at room temperature (298 K), presenting similar stress of 37.2–40.1 KPa at 99% compressive strain. Upon unloading, no residual strain has been observed, demonstrating the astonishing temperature-invariant super-elasticity of PI-10 aerogel. As a comparison, the elastic performances of polyurethane (PU) foam (20.2 mg/cm$^3$) and polyvinyl chloride (PVC) foam (23.3 mg/cm$^3$) were also estimated with compression of large deformation in liquid $N_2$. More than 90% plastic deformation was left in PU and PVC foam, while PI-10 aerogel perfectly recovered to its original shape, revealing the great advantage of covalently crosslinked PI-10 aerogels (Fig. 5c). To further confirm the temperature-invariant super-elastic performance, the storage modulus of PI-10 aerogel was investigated through a combination of direct measurement over 133 K to 573 K and calculation over 4 K to 132 K based on time-temperature equivalence theory[52,53]. As shown in Supplementary Fig. 19, the storage modulus at 4 K and 573 K was

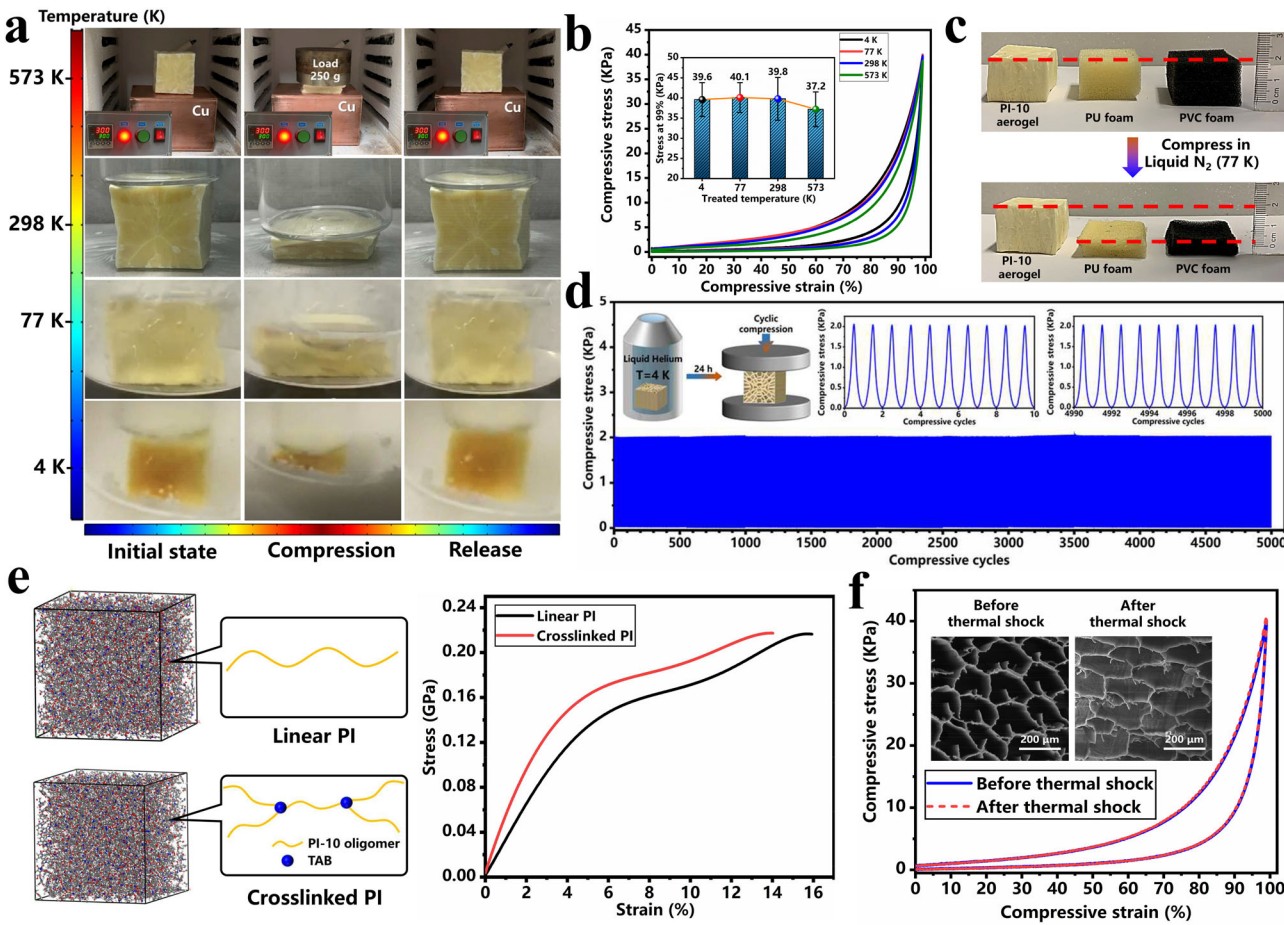

**Fig. 5 Mechanical performance of PI aerogel in various environments. a** Optical images of PI-10 aerogels during elastic tests at different temperatures from 4 K to 573 K. **b** Compressive stress-strain curves and compressive stress at 99% strain of PI-10 aerogels after being treated under different temperatures for 3 min. **c** Optical images of PI-10 aerogel, PU foam, and PVC foam before and after compression tests in liquid N₂. **d** Fatigue test curves of PI-10 aerogels after being treated in liquid helium (4 K) for 24 h. **e** Models and stress-strain curves of linear PI and crosslinked PI by simulation of molecular dynamics. **f** Compressive stress-strain curves and SEM images of PI-10 aerogels before and after thermal shock test.

0.72 MPa and 0.52 MPa, demonstrating a small variation between 4 K and 573 K, which indicates a satisfying temperature-invariant super-elasticity property. In addition, a fatigue test of the compressive mechanical property of the PI-10 aerogel treated in liquid helium (4 K) for 24 h also demonstrated that there was no significant deterioration of mechanical properties after long-term treatment at deep cryogenic temperatures (Fig. 5d). At deep cryogenic temperatures, polymer chains, chain segments, and even the secondary structures (rotation and stretch of covalent bonds) in polymer chains are frozen almost completely, resulting in a sharp increase in modulus and Poisson's ratio of bulk polymeric materials[54]. However, PU, PVC, and many other polymeric materials are born with low compressive strength. Extremely high modulus and low strength in a deep cryogenic environment easily give rise to structural fracture under stress. In this work, slight increases in bulk modulus, shear modulus, Young's modulus, and Poisson's ratio of constituent materials PI-10 were also observed by molecular dynamic simulations (Supplementary Fig. 20). Through the DMSO-FGFD process, chemical structure with covalent crosslinking and a framework structure with NPR behavior have been incorporated into PI aerogels, generating enhanced strength (Fig. 5e) matched with increased modulus and remitted energy impact from compression, endowing PI aerogels with excellent super-elasticity at deep cryogenic temperatures. In terms of the application environment with temperature jumps in aerospace, a rapid thermal shock

evaluation between 4 K and 573 K was also carried out on PI-10 aerogels (Fig. 5f and Supplementary Fig. 21). Before and after thermal shocks with a temperature jump of 569 K, PI-10 aerogel still maintains similar compressibility up to 99% strain and perfect recoverability, with no obvious structural damage being observed. This excellent resistance to thermal shock is vitally important for practical application in extreme environments in aerospace.

## Discussion

In summary, we have reported a DMSO-FGFD strategy to design and synthesize covalently crosslinked PI aerogels with super-elasticity. Benefiting from an innovative chemical imidization process, this PI aerogel exhibits remarkably ultralow volume shrinkage of 3.1% and an ultralow density of 6.1 mg/cm³, which are superior to reported elastic PI-based aerogels. Innovative mold design and temperature adjustment endowed the obtained PI aerogels with a radially distributed cellular structure to realize NPR of −0.2. Ultralow volume shrinkage and density, covalently crosslinked structure and NPR behavior endow the ultralight PI aerogels with the capacity to bear compressive strain up to 99% and perfectly recover their original shape. Surprisingly, obtained PI aerogels also exhibit marvelous super-elasticity at the deep cryogenic temperature of 4 K, which has never been achieved for any polymeric materials. Even after suffering from a thermal shock between 4 K and 573 K, PI aerogels still retain

compressibility up to 99% strain and perfect recoverability. To this end, these ultralight PI aerogels possess much promise for application as super-elastic materials resistant to deep cryogenic temperature in aerospace exploration.

## Methods

**Materials.** 4,4'-oxydianiline (ODA) (99.5%) and 4,4'-oxydiphthalic anhydrides (ODPA) (99.5%) were purchased from Changzhou Sunlight Pharmaceutical Co. Ltd. 1,3,5-Triaminophenoxybenzene (TAB) (99.5%) was provided by Haorui Chemicals Co., Ltd. Dimethyl sulfoxide (DMSO) was purchased from Shanghai Taitan Technology Co., Ltd and dried with molecular sieves prior to use. Acetic anhydride (AR) and triethylamine (AR) were purchased from Sinopharm Chemical Reagent Co., Ltd.

**Fabrication of covalently crosslinked PI aerogels.** Firstly, PI oligomers end-capped by anhydride were obtained through chemical imidization at room temperature by adding $n_1$ mol acetic anhydride and $n_1$ mol triethylamine into PAA precursors which were synthesized from $n_1$ mol ODA and $n_2$ mol ODPA in DMSO solvent. Subsequently, $n_{TAB}$ mol TAB was added into the DMSO solution containing PI oligomers to obtain a uniform mixture. In this work, to adjust the crosslinking degree, the relationship between $n_1$, $n_2$, and $n_{TAB}$ were designed as follows.

$$\frac{n_1}{n_2} = \frac{n}{n+1} \tag{2}$$

$$n_{TAB} = \frac{2}{3}(n_1 - n_2) \tag{3}$$

where $n$ ($n = 10, 20, 30, 40$) is the polymerization degree of PI oligomers, and the corresponding PI aerogels were marked as PI-n. Particularly, $n_1$ is equal to $n_2$ in preparing linear PI (PI-L) without crosslinkers. As an example, the preparation of PI-10 aerogels is described as follows: 133.5 g DMSO was added into a 250 mL three-necked flask equipped with a nitrogen inlet and a mechanical stirrer. 3.0036 g ODA (15 mmol) was added into the flask and dissolved completely, followed by adding 5.1186 g ODPA (16.5 mmol) into the solution. A PAA/DMSO solution with a solids content of 6 wt% was obtained after stirring for 12 h at room temperature. After that, PI oligomers were obtained by adding 3.0627 g (30 mmol) acetic anhydride and 3.3393 g (30 mmol) triethylamine into the PAA/DMSO solution and stirring for 1 h. The PI oligomers/DMSO solution was diluted into 0.5 wt% by adding more DMSO solvent, followed by adding 0.3994 g (1 mmol) TAB to obtain a uniform mixture of PI/TAB/DMSO. A directional freeze gelation process was carried out by adding the above-mentioned mixtures into the predesigned model on a freezing stage of −60 °C. After the solution was frozen entirely, the frozen gel was kept in the refrigerator for 24 h. And then, the sample was freeze-dried for 4 days in a freeze dryer with temperatures of −110 °C in a cold trap and −3 °C in the sample chamber, and the pressure was kept at 1 Pa. The dried samples were treated at 250 °C in a vacuum oven for 3 h to obtain the final PI-10 aerogels.

**Characterizations.** In situ observations of the pre-freezing and freeze-drying process were carried out on LINKAM FDCS196 Microscopy System of Vacuum Freeze-drying. Briefly, 2 μL solution was added on the testing stage which was frozen to −60 °C by 5 °C/min and then heated to −2 °C by 1 °C/min. The pressure was kept at 101 KPa in the freezing process and 5 Pa in the heating process. Attenuated total reflectance infrared spectroscopy (ATR-FTIR) was recorded on a Nicolet is10 spectroscope with the range of 4000–600 cm$^{-1}$ by averaging 32 scans. The microstructure of the aerogels was observed on a scanning electron microscope (SEM) (TESCAN MAIA3) at an accelerating voltage of 15 KV, and the wall thickness and pore size of the aerogels was analyzed using MAIA TC software. Differential scanning calorimetry (DSC) was performed on a Netzsch DSC 404F3 at a scan rate of 10 °C/min in flowing nitrogen. The thermal conversion process was analyzed by Netzsch TG 209 F1 Thermogravimetric Analyzer (TGA) at a heating rate of 10 °C/min in flowing nitrogen. The molecular weights of PI-L, as well as the PI oligomers with polymerizations degree of 40, 30, 20, and 10, were characterized by gel permeation chromatography (GPC, Agilent 1260 Infinity) with a mobile phase of dimethyl formamide (DMF), the flow rate of 1 mL/min, and injective volume of 20 μL. The concentration of the PI/DMF solution was 1 mg/mL. Rheological behavior measurements were performed on a HAAKE MARS III Rotational Rheometer at 25 °C. The shearing rate was kept constant at 10 rad/s for the crosslinking process analysis of PI oligomers and TAB. An increasing shearing rate from 0 to 300 rad/s was taken to analyze the polymerization degree of PI oligomers at 25 °C. The compressive tests of PI aerogels were performed on an Instron 5966 material testing instrument, 5 parallel tests were performed on each series of samples. The strain ramp rate was kept at 10 mm/min for all compressive tests, with the size of the samples of (L)30 mm × (W)30 mm × (H)30 mm. The fatigue tests were performed on a TA ElectroForce 3220 Mechanical Test Instrument with a compressive frequency of 1 Hz and compressive strain of 50%, and the size of the samples was (L)8 mm × (W)8 mm × (H)8 mm. The storage modulus of PI-10 aerogel between 133 K and 573 K was directly measured on the METTLER DMA861 instrument under an oscillatory of ε = ±5% at 1 Hz with a heating rate of 5 K/min in an N$_2$ atmosphere. The storage modulus of PI-10 aerogel between 4 K and 132 K was calculated based on the time-temperature equivalence theory by relating the storage modulus measured at 5 Hz, 100 Hz, and 500 Hz

between 132 K and 298 K to those between 4 K and 132 K by a shift of the modulus-time curves along the time axis.

**Molecular dynamics (MD) simulations.** The MD simulations for Fraction of Free Volume (FFV) and mechanical properties were carried out based on Material Studio 2019 software. In the calculation of FFV, 5 polymeric chains with a polymerization degree of 20 were packed into a periodic cube for the construction of an amorphous cell followed by a dynamic optimization in the Forcite module with the force field of COMPASS. NPT (Number of atoms, pressure, and temperature are constant) and NVT (Number of atoms, volume, and constant temperature are constant) ensembles were taken to deduce the final equilibrium model. And the temperature and pressure in the equilibrium process of amorphous cells were controlled by the Nose thermostat and Berendsen barostat. FFV $= \frac{V_f}{V_{sp}}$, where the free volume was calculated by $V_f = V_{sp} - 1.3V_w$, using geometric measures of the specific volume ($V_{sp}$), and van der Waals volume ($V_w$). In the calculation of stress-strain curves, 10 crosslinked PI chains were packed into a periodic cube for the construction of an amorphous cell followed by a dynamic optimization in Forcite module with the force field of COMPASS. NPT and NVT ensembles were taken to deduce the final equilibrium model. And the temperature and pressure in the equilibrium process of the amorphous cells were controlled by Andersen's thermostat and Berendsen barostat. And then, a uniaxial tensile test was carried out on the constructed amorphous cell with a strain rate of $2 \times 10^8 \, s^{-1}$ on the $z$ direction at 300 K until the maximum strain of 17% in an NPT ensemble. In the calculation of modulus, 10 crosslinked PI chains were packed into a periodic cube for the construction of an amorphous cell followed by a dynamic optimization in the Forcite module with the force field of COMPASS. NPT and NVT ensembles were taken to deduce the final equilibrium model at 4 K and 298 K. And the temperature and pressure in the equilibrium process of the amorphous cells were controlled by a Nose thermostat, and Berendsen barostat, respectively.

**Simulation of physical processes.** All the simulations of physical processes in this work are implemented by the finite element method (FEM) with the COMSOL Multiphysics software. In the simulation of the freeze-drying process, a 3D model with a size of (L)30 mm × (W)30 mm × (H)30 mm was created, and the volume fraction of DMSO is 99.5%. The Physical field is the Transport of Concentrated Species. In the simulation of temperature distribution on the bottom of the mold, a 2D model with (L)30 mm × (W)30 mm was created, and the physical field is Solid Heat Transmission in which the initial temperatures of surrounding and center are −60 °C and 25 °C. A 2D model with (L)30 mm × (W)30 mm and a 3D model with 2 hexagon frameworks next to each other with a wall thickness of 2 μm was created to simulate the NPR behavior and the deformation process of cellular structure in PI-10 aerogels, and the physical field in Solid Mechanics.

The authors affirm that human research participants provided informed consent for publication of the images in Supplementary Fig. 18.

## Data availability

All data needed to evaluate the conclusions in the paper are presented in the paper and/or the Supplementary Materials. Additional data related to this paper may be requested from the authors.

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

## Acknowledgements

This work was financially supported by the National Natural Science Foundation of China (51972064).

## Author contributions

Y.C. performed most of the tests and analyses and wrote the manuscript. X.Z. and P.D. came up with constructive proposals and revised the manuscript. Y.Q performed mechanical properties and molecular weight characterizations. W.Y. helped to modify the experiments. J.M. and P.M.A. revised the manuscript and came up with significant suggestions. J.S. and M.Y. supervised all research phases. All authors discussed and commented on the manuscript.

## Competing interests

The authors declare no competing interests.
