## [Peer Review File · Nature Communications]

Reviewers' Comments:

Reviewer #1:

Remarks to the Author:

I have reviewed the manuscript (NCOMMS-21-01556) entitled as "Super-elasticity at 4K of Covalently Crosslinked Polyimide Aerogels with Ultrahigh Negative Poisson's Ratio". The authors fabricated covalently crosslinked polyimide aerogels through directional freezing of PI oligomers/DMSO solution containing TAB crosslinkers, followed by freeze drying. The obtained PI aerogels consisted of parallel hollow tubes along growing direction of DMSO crystals. The covalently crosslinked chemical structure, cellular architecture, negative Poisson's ratio (-0.2) and ultralow density (6.1 mg/cm³) endow the PI aerogels with excellent elasticity. An elastic compressive strain up to 99% even in liquid helium (4K) could be achieved. These results are novel and meaningful. The experimental procedure would be valuable to other researchers in the related field. In my opinions, this manuscript could be accepted after some revisions. Some comments or suggestions were listed below:

- 1)The molecular weight or polymerization degree of linear PI (PI-L) should be provided;
- 2)In Figure 1(a), the solid content of PI solution should be given; In Figure 1(d), the polymerization degree of PI oligomer should be given; the same problems were in Figure 1(d) and (e);
- 3)In Figure 2, the solid content or polymerization degree of the samples should be also provided;
- 4)In Figure 2(d), the SEM photos, the xy view and yz view are puzzling. I suggested to describe them as viewing along z direction (z view) and viewing along x or y direction (x, y view);
- 5)In Figure 3, giving the density of samples are helpful to understand the differences of compressive performances. The compression direction should be also explained (vertical channel direction);
- 6)In Figure 4, the density of samples including PU, PVC foams should be provided.

Reviewer #2:

Remarks to the Author:

Interesting work, but lacking a lot of details. Authors are asked to revise the manuscript.

1. The term "freeze-gelling" is incorrect and confusing. The correct word for transformation of a sol into a gel is "gelation".
2. The process that the authors used to obtain the so called "aerogel" produces indeed a foam-like material with high degree of directionality induced by the directional freeze drying process. The pores are a few tens of micrometer in diameter. The classical aerogel materials (about 100 years old field) always contain mesopores. These authors did not show if their materials contained any mesopores. Also the authors did not show if the skin layers of this foam materials were porous or solid precipitates? If latter, then the authors need to explain this behavior. If former, the authors need to have high resolution SEM images to show mesopores.
3. The structural resilience and the negative Poisson's ratio indicate that the "skin layers" were indeed filled with fine strands of PI, thus allowed organization of PI domains without undergoing fracture.
4. It is important to show storage modulus of the PI material at 4K and 573K.
5. I do not believe the authors could avoid thick skin layers at the top and bottom of the specimens. Were these skin layers removed before making measurements? If not, please explain how thick such layers were and what impact these had on stress vs. strain behavior.

The point-to-point responses to reviewers' comments

Dear editor and reviewers,

This work was submitted to “Nature Communications” with the manuscript number of **NCOMMS-21-01556**. We appreciate the editor for giving us an opportunity to revise the manuscript, and also appreciate the reviewers for giving us constructive and high-quality suggestions to improve our work furtherly. After careful evaluation of the manuscript and reviewers' comments, complementary experiments and characterizations are adopted with detailed discussion. We hope the revised paper could meet the approval of both editor and reviewers.

The authors of this work have discussed on this paper carefully and have taken into the accounts of the reviewers. The manuscript is revised accordingly and marked with **a highlight** where changes have been made. We have also paid special attention to the “Editor's comments” and make sure that our manuscript is consistent with the requirements of this journal. We ardently hope that in light of our responses to all the comments, the reviewers and readers can reach a more favorable view with the revised manuscript. We believe this work is suitable for a broad scope of readers and suitable for such a flagship journal as the “Nature Communications”.

Please find point-to-point responses to reviewers' comments attached below.

Sincerely yours,

Mingxin Ye, Jianfeng Shen

Institute of special materials and technology

Fudan University, Shanghai, China

Email: mxye@fudan.edu.cn, jfshen@fudan.edu.cn

Fax: 021-55664095

Tel: 021-55664095

To reviewer #1:

General comments. These results are novel and meaningful. The experimental procedure would be valuable to other researchers in the related field. In my opinion, this manuscript could be accepted after some revisions. Some comments or suggestions were listed below.

General answers. We really appreciate the positive comments and highly valuable suggestions from the reviewer. These suggestions are valuable and helpful for revising and improving our paper, as well as the important guiding significance for our research. We have tried our best to revise the manuscript according to these comments and suggestions.

Q1. The molecular weight or polymerization degree of linear PI (PI-L) should be provided.

A1. Thanks for the constructive suggestion. We have characterized the molecular weight of PI-L, as well as other PI oligomers with a polymerization degree of 40, 30, 20, and 10, by gel permeation chromatography (GPC). These results were shown in **Supplement Figure 3** and analyzed with blue words on page 5. The characterization methods were revised with blue words in the section of methods on page 16.

(1) Page 5

“The molecular weights of oligomers were further investigated with gel permeation chromatography (GPC), and the results were shown in **Supplementary Figure 3**. It demonstrates that the number average polymerization degree (\bar{X}_n) of PI-L can reach 98, and \bar{X}_n of the other oligomers are consistent with the experimental design (10,20,30,40), which also accords with viscosity test results.”

(2) Page 16

“The molecular weights of PI-L, as well as the PI oligomers with polymerizations degree of 40, 30, 20, and 10 were characterized by gel permeation chromatography (GPC, Agilent 1260 Infinity) with a mobile phase of dimethyl formamide (DMF), flow rate of 1 mL/min, and injective volume of 20 μ L. The concentration of the

PI/DMF solution was 1mg/mL.”

(3) Supplementary Materials

Supplementary Figure 3. Molecular weight and number-average polymerization degree of PI oligomers with various polymerization degrees.

Q2. In Figure 1(a), the solid content of PI solution should be given; In Figure 1(d), the polymerization degree of PI oligomer should be given; the same problems were in Figure 1(d) and (e).

A2. According to the suggestion of the reviewer, the solid content of PI solution in Figure 1(a), the polymerization degree of PI oligomer in Figure 1(d), and both the solid content and polymerization degree in Figure 1(e) have been provided. We have revised the particular parts with blue words on page 5 and page 6, as well as the figure captions on page 24.

(1) Page 5

“To explore the process of gelling interaction between PI oligomers and TAB crosslinkers, the rheological behavior of PI/TAB/DMSO mixtures **with solid content of 6.0 wt%** was observed on a rotational rheometer as shown in **Figure 1a.**”

(2) Page 6

“The Differential Scanning Calorimetry (DSC) curves demonstrated the melting temperature range of DMSO crystals containing different amounts of PI oligomers **with a polymerization degree of 10 (Figure 1d).**”

“Furthermore, DMSO solvent containing linear PI with a solid content of 0.5 wt% was chosen to in-situ observe the formation of DMSO crystals in the pre-freeze and analyze temperature range of vacuum drying by the vacuum freeze-drying microscopy system. As shown in **Figure 1e**, when cooled down from 25 °C to -60 °C at ambient pressure, parallel DMSO crystals come into being as templates to push PI chains to aggregate between them, demonstrating that the DMSO solvent was beneficial to prepare PI aerogels with well-organized directional frameworks.”

(3) Page 24

“**Figure 1. Fabrication process of covalently cross-linked PI aerogel with DMSO solvent.** (a) Shearing viscosity curves of 6 wt% PI oligomers solutions with different contents of TAB cross-linker under a constant shearing rate. (b) Mechanism of the directional freeze gelation process. (c) Optical images after thawing frozen gel of PI/TAB/DMSO with a solid content of 0.5 wt%. (d) DSC curves of DMSO solvent containing different content of PI oligomers with polymerization degree of 10. (e) Microscopy images of the pre-freezing and freeze-drying process from the microscopy system of vacuum freeze-drying based on PIM-10 with a 0.5 wt% solid content in DMSO solvent. (f) Simulation of residual DMSO during freeze-drying process by finite element analysis.”

Q3. In Figure 2, the solid content or polymerization degree of the samples should be also provided.

A3. Thanks for the suggestion of the reviewer. The investigations of structures and morphologies were based on the PI aerogel freeze-dried from PI/TAB/DMSO solution with solid content of 0.5 wt%. According to the suggestion, we have revised the particular parts with blue words on page 7.

(1) Page 7

“The PI aerogels freeze-dried from 0.5 wt% PI/TAB/DMSO solution has been chosen to investigate the structure and morphology of PI aerogels with NPR behavior.”

Q4. In Figure 2(d), the SEM photos, the xy view and yz view are puzzling. I

suggested to describe them as viewing along z direction (z view) and viewing along x or y direction (x, y view).

A4. The suggestions of the reviewer make the descriptions much clearer to be understood by audiences. We have revised the related descriptions with blue words on page 8, and marks in Figure 2d as well as the figure captions on page 25.

(1) Page 8

“As demonstrated by scanning electron microscope (SEM) images (**Figure 2d**) viewing along z direction (z view) and viewing along x or y direction (x, y view), PI-10 aerogel consists of parallel hollow tubes with a pore size of 200~300 μm and a wall thickness of ~ 2 μm , which is the result of evaporation of DMSO crystals in vacuum freeze-dryer.”

(2) Figure 2d

(3) Page 25

“**Figure 2. Structure and morphology of PI aerogels.** (a) DSC curves of PI aerogels. (b) Average shrinkage rates of PI aerogels from chemical imidization, thermal imidization and related references, 5 parallel tests were performed on each series of samples. (c) Optical images of the volume and weight of a typical PI aerogel. (d) SEM images viewing along z direction (z view) and along x direction (x view) of anisotropic PI-10 aerogels. (e) Temperature distribution and growing direction of DMSO crystals at the bottom of mold. (f) overview of radial distributed morphology in PI-10 aerogels. (g) Simulated NPR behavior of PI aerogels. (h) Sequential optical images showing NPR under compression vertical channel direction.”

Q5. In Figure 3, giving the density of samples are helpful to understand the differences of compressive performances. The compression direction should be also

explained (vertical channel direction).

A5. Thanks for the suggestions of the reviewer. According to the suggestions, the density of PI aerogels has been provided based on the results in **Supplementary Figure 9**, and the compressive direction has been explained. We have revised the particular parts with blue words on page 10, 11 to 12, and Figure captions on page 26.

Supplementary Figure 9. Density and porosity of PI aerogels prepared from PI/TAB/DMSO with different solid contents.

(1) Page 10

“Interestingly, the bulk PI-10 aerogel with a low density of 6.1 mg/cm³ is able to bear 2000 times of its own weight (**Supplementary Figure 10**) along channel direction, which clearly reveals its strong stiffness.”

“As shown in **Figure 3b**, all prepared PI aerogels with ultra-low densities of 6.1 to 8.0 mg/cm³ (**Supplementary Figure 8**) can recover under 70% compressive strain vertical channel direction, and exhibit a growing tendency of stress with increasing crosslinking degree. Covalently crosslinked structure endows PI-10 aerogels with a compressive stress of 6.5 kPa under 70% strain vertical channel direction, which is 2.5 times that of PI-L without crosslinking.”

“The compressibility and elasticity vertical channel direction of PI aerogels have been further evaluated under an ultimate strain of 99%. As shown in **Figure 3c**, all PI aerogels with different crosslinking degrees can be compressed to 99% due to their extraordinary flexibility and ultra-low densities of 6.1 to 8.0 mg/cm³.”

“The highly recoverable compressibility vertical channel direction of PI-10 aerogels

can be mainly attributed to the enhanced mechanical properties because of their covalently crosslinked structure and NPR behavior via of their radially distributed cellular structure.”

(2) Page 11 to 12

“A cyclic compression vertical channel direction with a sinusoidal frequency of 1 Hz test was carried out to estimate the mechanical durability of PI-10 aerogels with a density of 6.1 mg/cm^3 .”

(3) Page 26

“**Figure 3. Mechanical performance of PI aerogels.** (a) Optical images of bending to 180° and compressing to 99% strain vertical channel direction of PI-10 aerogel with a density of 6.1 mg/cm^3 . (b) Compressive stress-strain curves with 70% strain vertical channel direction of PI aerogels with different crosslinking degrees. (c) Compressive stress-strain curves with 99% strain and residual strain vertical channel direction of PI aerogels with different crosslinking degrees. (d) Comparison of ultimate recoverable strains and densities of PI-10 aerogels with reported polymeric aerogels. (e) Simulated results of local stress distribution in cellular PI aerogels. (f) In-situ sequential SEM images of microstructure in PI-10 aerogels with different strain. (g) Fatigue test curves on vertical channel direction and SEM images of PI-10 aerogels before and after fatigue test of 5000 cycles.”

Q6. In Figure 4, the density of samples including PU, PVC foams should be provided.

A6. Thanks for the suggestions of the reviewer. According to the suggestions, the density of commercial PU foams (20.2 mg/cm^3) and PVC foams (23.3 mg/cm^3) have been provided, and we have revised the particular parts with blue words on page 13.

(1) Page 13

“As a comparison, the elastic performances of polyurethane (PU) foam (20.2 mg/cm^3) and polyvinyl chloride (PVC) foam (23.3 mg/cm^3) were also estimated with compression of large deformation in liquid N_2 .”

To reviewer #2:

General comments. Interesting work, but lacking a lot of details. Authors are asked to revise the manuscript.

General answers. We appreciate the reviewer very much for the encouraging comments and constructive suggestions. These comments are valuable and helpful for revising and improving our paper, as well as the important guiding significance for our research. We have tried our best to revise our manuscript based on these comments and suggestions.

Q1. The term "freeze gelling" is incorrect and confusing. The correct word for transformation of a sol into a gel is "gelation".

A1. Thanks for providing a more accurate description of the sol-gel transformation in our study.

The term “freeze-gelling” has been changed into “freeze gelation”, which is consistent with the descriptions in the previous literature. (Lin, Y. et al. *Adv. Mater.* **2016**, 36, 7993; Hess, U. et al. *Mater. Sci. Eng. C* **2016**, 67 542; Qasim, S. B., et al. *Acta Biomater.* **2015**, 23, 317.).

We have revised the particular parts with blue words on page 1, 3, 4, 5, 6, 9, 16, 24.

(1) Page 1

“In this work, super-elasticity at deep cryogenic temperature of covalently crosslinked polyimide (PI) aerogels is achieved based on scalable and low-cost directional dimethyl sulfoxide crystals assisted **freeze gelation** and freeze-drying strategy.”

(2) Page 3

“In this study, we proposed a novel directional dimethyl sulfoxide crystal assisted **freeze gelation** and freeze-drying (DMSO-FGFD) strategy to construct covalently crosslinked PI aerogels with super-elasticity at deep cryogenic temperatures even down to 4 K.”

(3) Page 4

“At the initial **freeze gelation** stage, the DMSO crystals grew horizontally from the

periphery to the center, resulting in radially distributed crystals due to a predesigned model and temperature adjustment.”

(4) Page 5

“**Figure 1b** illustrates the mechanism of the freeze gelation process.”

(5) Page 6

“After thawing at 35 °C, the frozen gel with TAB transformed into an agglomerate wet gel (**Figure 1c**), verifying the formation of PI with covalent crosslinking in the freeze gelation process, while the frozen gel without TAB returned to a flowing state in contrast (**Supplementary Figure 4**).”

(6) Page 9

“With the help of finite element simulation based on COMSOL Multiphysics software, a radial temperature distribution (**Figure 2e**) was achieved at the initial stage of freeze gelation through a design with a slightly sunken center on the outer bottom of the mold (**Supplementary Figure 9**).”

(7) Page 16

“A directional freeze gelation process was carried out by adding above-mentioned mixtures into the predesigned model on a freezing stage of -60 °C.”

(8) Page 24

“**Figure 1**. (b) Mechanism of the directional freeze gelation process.....”

Q2. The process that the authors used to obtain the so called "aerogel" produces indeed a foam-like material with high degree of directionality induced by the directional freeze-drying process. The pores are a few tens of micrometer in diameter. The classical aerogel materials (about 100 years old field) always contain mesopores. These authors did not show if their materials contained any mesopores. Also, the authors did not show if the skin layers of this foam materials were porous or solid precipitates? If latter, then the authors need to explain this behavior. If former, the authors need to have high resolution SEM images to show mesopores.

A2. Thanks for the comments and suggestions of the reviewer.

(1) We have referred to more literatures on this issue, thus trying our best to make the statement concise. Indeed, some researchers would like to call such kinds of porous materials as foams or sponges, but it is still controversial. The term of aerogel was first introduced by Kistler in 1931 to designate gels in which the liquid was replaced with gas, without collapsing the gel solid network. (Kistler, S. *Nature* **1931**, 127, 741.) By considering recent developments in the fabrication of aerogels, it is indeed more realistic to define these materials with reference to the initial idea of Kistler, simply as gels in which the liquid has been replaced by air, with very moderate shrinkage of the solid network, which is a generalized definition of aerogel. (M. A. Aegerter, et al, *Aerogels Handbook Part I*, Springer New York, 2011.)

Our covalently crosslinking PI aerogels were fabricated via a sol-gel method followed by freeze-drying process to replace the DMSO solvent with air, and the final shrinkage is as low as 3.1%. Obviously, it meets all the features in the definition. In addition, mesopores could also be observed on the walls of cellular architecture as shown in **Supplementary Figure 8b**, which further proved the rationality of defining our porous PI as aerogel. After careful consideration based on these literatures and supplementary results, we thought it was appropriate to call our PI material aerogel.

(2) The skin layers mentioned in the comments of the reviewer come into being at the periphery and bottom of PI aerogel. According to the suggestion of reviewer, we have further investigated the structure and morphology of skin layer via SEM. **Supplementary Figure 8a** and **Supplementary Figure 8b** show the morphology of cross section and surface of the skin layer around the PI-10 aerogel. Obviously, the skin layer is a well-organized channel structure with some nanosized pores, which is similar to the major architecture, revealing that the skin layer should be a horizontally periodic spread of the unidirectional architectures instead of a dense layer. **Supplementary Figure 8c** and **Supplementary Figure 8d** display the cross section and surface of the skin layer at the bottom of PI-10 aerogel. Although the skin layer at the bottom is denser than the major architecture, the skin layer is still a porous structure instead of a dense layer. However, the sizes of the pores ($\approx 10\sim 20\ \mu\text{m}$) at the

bottom layer are much smaller than those in major architecture (200~300 μm), which results in a visually dense layer with naked eyes. The unidirectional freeze-gelation was carried out in a mold containing uniform PI/TAB/DMSO mixture put on a freezing plate of $-60\text{ }^{\circ}\text{C}$. The freezing temperature at the bottom of the mold is close to the temperature of the freezing plate ($-60\text{ }^{\circ}\text{C}$), which is much lower than the freeze point of DMSO resulting in the formation of smaller sized DMSO crystals as a very fast freezing rate. Thus, smaller sized pores formed at the bottom of PI aerogel after freeze-drying.

(3) We have added more discussion on these issues in Supplementary Materials and revised the particular parts on page 8 to 9 with blue words.

(1) Page 8 to 9

“Additionally, in view of the effect of mold inner walls and the lower freeze temperature at the mold bottom, the morphology of skin layers at the periphery and bottom of PI aerogel has also been investigated in **Supplementary Figure 8**. The skin layer around the aerogel is a well-organized channel structure which is similar to the major architecture, revealing that the skin layer should be a horizontally periodic spread of the unidirectional architectures, but the bottom skin layer displays smaller sized pores (10~20 μm).”

(2) Supplementary Materials

Supplementary Figure 8. Morphology of skin layers. (a) Cross-sectional SEM images of the skin layer around the aerogel. (b) Surface SEM images of the skin layer around the aerogel. (c) Cross-sectional SEM images of the skin layer at the bottom of the aerogel. (d) Surface SEM images of the skin layer at the bottom of the aerogel.

Q3. The structural resilience and the negative Poisson's ratio indicate that the "skin layers" were indeed filled with fine strands of PI, thus allowed organization of PI domains without undergoing fracture.

A3. Thanks for the comments of the reviewer.

As shown in **Figure R1 (a)** and **Figure R1 (b)**, no obvious differences were visually observed on the surface morphology around the aerogel before and after removing the skin layer, but a larger porous surface could be observed after removing the bottom skin layer. Through the comparison of Supplementary Figure 8a, Supplementary Figure 8b and Figure 2(d) in **Figure R2**, the surface morphology around the aerogel before and after removing the skin layer is almost similar, but the size of the pores on

skin layer at the bottom is much smaller compared with the major architecture. In addition, after removing both surrounding and bottom skin layers, the PI aerogel still displays obvious NPR behavior as shown in **Figure R3**. Therefore, we have to accept that the skin layer at the bottom is capable of improving the resilience and protecting the major architecture from fracture in some degree owing to the relative denser porous structure, which has been further discussed in **Q5** and **A5**.

Figure R1. (a) The optical images of surface morphologies around the aerogel before and after removing the skin layer. (b) The optical images of surface morphologies before and after removing the skin layer at the bottom.

Figure 2d. SEM images of major architectures in PI aerogel

Supplementary Figure 8a. Cross-sectional SEM images of the skin layer around the aerogel.

Supplementary Figure 8b. Surface SEM images of the skin layer around the aerogel.

Figure R2. Comparison of Supplementary Figure 8a, Supplementary Figure 8b and Figure 2(d).

Figure R3. NPR behavior of PI aerogel after removing skin layers.

Q4. It is important to show storage modulus of the PI material at 4K and 573K.

A4. Thanks for the constructive suggestions of the reviewer.

The storage modulus is an important parameter for the elastic materials. We have tried our best to search for instruments for this characterization. However, there isn't any commercial DMA instrument which can measure the storage modulus at such a low temperature down to 4 K according to our investigations. Therefore, we directly measured the storage modulus of PI-10 aerogel between 133 K (which is the limit of the detection temperature with the most advanced DMA instruments we know at present) and 573 K. As we all know, both the modulus and the compliance of viscoelastic materials demonstrate time dependent quantities. Therefore, we approximately calculated the storage modulus of PI-10 aerogels between 4 K and 133 K based on the time-temperature equivalence theory (G. M. Swallowe, *Mechanical Properties and Testing of Polymers: An A-Z Reference*, Ed. Springer Netherlands, Dordrecht, **1999**, chapter 54, page 249-251; Dorléans V, et al. A Viscoelastic-Viscoplastic Time-Temperature Equivalence for Thermoplastics. *12th European LS-DYNA Conference at Koblenz, Germany*; **2019**). The storage modulus of PI-10 aerogel between 4 K and 133 K was calculated through relating the storage modulus measured at 5Hz, 100 Hz, and 500 Hz between 133 K and 298 K to those between 4 K and 133 K by a shift of the modulus-time curves along the time axis.

The obtained storage modulus-time curves were shown in Supplementary Materials. Red curve was obtained from calculation based on time-temperature equivalence theory and the black curve was obtained from direct measurement. More analysis and testing method of the storage modulus was supplemented on page 13 and

17 with blue words.

(1) Supplementary Materials

Supplementary Figure 19. Storage modulus of PI-10 aerogel between 4 K and 573 K.

(2) Page 13

To further confirm the temperature-invariant super-elastic performance, the storage modulus of PI-10 aerogel was investigated through a combination of direct measurement over 133 K to 573 K and calculation over 4 K to 132 K based on time-temperature equivalence theory^{53,54}. As shown in **Supplementary Figure 19**, the storage modulus at 4 K and 573 K was 0.72 MPa and 0.52 MPa, demonstrating small variation between 4 K and 573 K, which indicates a satisfying temperature-invariant super-elasticity property.

(3) Page 17

The storage modulus of PI-10 aerogel between 133 K and 573 K was directly measured on the METTLER DMA861 instrument under an oscillatory of $\varepsilon = \pm 5\%$ at 1 Hz with a heating rate of 5 K/min in a N_2 atmosphere. The storage modulus of PI-10 aerogel between 4 K and 132 K was calculated based on the time-temperature equivalence theory through relating the storage modulus measured at 5 Hz, 100 Hz, and 500 Hz between 132 K and 298 K to those between 4 K and 132 K by a shift of the modulus-time curves along the time axis.

Q5. I do not believe the authors could avoid thick skin layers at the top and bottom of the specimens. Were these skin layers removed before making measurements? If not, please explain how thick such layers were and what impact these had on stress vs. strain behavior.

A5. Thanks for the suggestion of the reviewer. While testing the mechanical performance, we used the as-prepared sample without removing these skin layers before measurements. According to the suggestion of the reviewer, we have carried out a comparison test of mechanical performance of PI-10 aerogel with and without all skin layers, bottom skin layer, and surrounding skin layer to investigate the effect of skin layers on compressive stress and resilient strain.

(1) As shown in **Supplementary Figure 13a**, after removing all the skin layers (both surrounding and bottom skin layers), the compressive stress decreased about 3.5% (from 39.7 KPa to 38.3 KPa at 99% strain), which is close to that after removing the bottom skin layer (**Supplementary Figure 13b**). However, when the surrounding skin layer was removed, no obvious deterioration of mechanical performance was detected (**Supplementary Figure 13c**). It illustrates that the bottom skin layer is capable of improving mechanical performance owing to its smaller sized pores (more pore walls of PI) compared with the major architectures, but the surrounding skin layer has little effect on mechanical performance as it is an expansion of major architecture instead of a dense layer.

(2) In addition, through the analysis of magnifying in stress-strain curves with strain between 0~15%, we find that almost all the PI-10 aerogels can spring back to their original shape during release, whether they were removed skin layers or not. It demonstrated little impact of skin layers on ultimate resilient strain, which mainly depends on the major cellular architectures with radial distribution and mechanical properties of covalently cross-linked PI.

(3) We have added more discussion on these issues based on these characterizations and revised the particular parts on page 10 and Supplementary Materials with blue

words.

(1) Page 10

“Additionally, in view of the effect of skin layers at the periphery and bottom of PI aerogel, a comparison investigation has been carried out in **Supplementary Figure 13**. It demonstrated minor improvement of compressive stress but little impact on ultimate resilient strain of skin layers.”

(2) Supplementary Materials

Supplementary Figure 13. Comparison investigation on stress and strain before and after removing skin layers. (a) Stress-strain curves for PI-10 aerogel with and without all skin layers. (b) Stress-strain curves for PI-10 aerogel with and without bottom skin layer. (c) Stress-strain curves for PI-10 aerogel with and without surrounding skin layer.

Reviewers' Comments:

Reviewer #1:

Remarks to the Author:

The authors have revised their manuscript according to my comments and suggestions. I recommend it for publication.

Reviewer #2:

Remarks to the Author:

The authors tried their best to respond to my comments. Some of their responses backed by new measurements are logical. I still have questions on the existence of pores on the walls between the adjoining cells and why the authors could brand their materials as aerogels. However, at this point, I believe the work can be accepted.

To reviewer #1:

General comments. The authors have revised their manuscript according to my comments and suggestions. I recommend it for publication.

General answers. We really appreciate your recommendation for publication of our research article in *Nature Communications*.

To reviewer #2:

General comments. The authors tried their best to respond to my comments. Some of their responses backed by new measurements are logical. I still have questions on the existence of pores on the walls between the adjoining cells and why the authors could brand their materials as aerogels. However, at this point, I believe the work can be accepted.

General answers. We really appreciate your positive comments on our revised manuscript and recommendation for its publication in *Nature Communications*.

The existence of pores on the walls between the adjoining cells may be attributed to the sublimation of the nano/micro branch DMSO crystals embedded in the walls. The walls between the adjoining cells form based on volume exclusion effect during unidirectional freeze gelation process in which the PI oligomers with TAB were expelled to the boundaries of the DMSO crystals because of the volume exclusion effect. However, DMSO crystals do not always grow like smooth pillars, but some micro/nano branch DMSO crystals can form around the vertically growing DMSO crystals. (Deville S, et al. *Science*, 311, 515-518 (2006).). These nano/micro branch DMSO crystals are inlaid in the crosslinked PI walls during the volume exclusion process. Thus, some mesoporous pores can form on the walls as the sublimation of the branch DMSO crystals during the freeze-drying process.

From the point of aerogel definition in classical books, our covalently crosslinking PI aerogels were fabricated via a sol-gel method followed by freeze-drying process to replace the DMSO solvent with air, and the final shrinkage is as low as 3.1%. Obviously, it meets the aerogel definition that gels in which the liquid has been replaced by air, with very moderate shrinkage of the solid network. (M. A. Aegerter, et al, *Aerogels Handbook Part I*, Springer New York, (2011).). Besides, from the perspective of

fabrication process, our porous covalently crosslinked PI materials were fabricated by strictly sol-gel method, which was the classical method for aerogel fabrications. Additionally, all the physical parameters, such as density (6.1 mg/cm^3) and porosity (99.57%), are obviously better than general foam or sponge materials. Furthermore, from the point of chemical structure, our super-elastic PI materials possess covalently crosslinked network structure, which is same with reported PI aerogels. (Li X, et al. *ACS Nano*, (2021); Vivod SL, et al. *ACS Appl. Mater. Interfaces*, 12, 8622-8633 (2020).). After careful consideration based on these literatures and supplementary results, we thought it was appropriate to call our PI material aerogel.

All in all, we really appreciate your recommendation for publication of our research article in *Nature Communications*. We believe it will be interesting to the readers of *Nature Communications*.